# The impact of family environment on self-esteem and symptoms in early psychosis

**Lídia Hinojosa-Marqués**[1☯], **Manel Monsonet**[1☯], **Thomas R. Kwapil**[2], **Neus Barrantes-Vidal**[1,3,4]*

**1** Departament de Psicologia Clínica i de la Salut, Universitat Autònoma de Barcelona, Barcelona, Spain, **2** Department of Psychology, University of Illinois at Urbana-Champaign, Champaign, IL, United States of America, **3** Sant Pere Claver- Fundació Sanitària, Barcelona, Spain, **4** Centre for Biomedical Research on Mental Health (CIBERSAM), Madrid, Spain

☯ These authors contributed equally to this work.
* neus.barrantes@uab.cat

**Data Availability Statement:** The authors of the present study confirm that some access restrictions apply to the data underlying the findings. The consent form that participants signed before participating in the study, approved by the

## Abstract

Expressed emotion (EE) and self-esteem (SE) have been implicated in the onset and development of paranoia and positive symptoms of psychosis. However, the impact of EE on patients' SE and ultimately on symptoms in the early stages of psychosis is still not fully understood. The main objectives of this study were to examine whether: (1) patients' SE mediated the effect of relatives' EE on patients' positive symptoms and paranoia; (2) patients' perceived EE mediated the effect of relatives' EE on patients' SE; (3) patients' SE mediated between patients' perceived EE and patients' symptomatology; and (4) patients' perceived EE and patients' SE serially mediated the effect of relatives' EE on patients' positive symptoms and paranoia. Incipient psychosis patients (at-risk mental states and first-episode of psychosis) and their respective relatives completed measures of EE, SE, and symptoms. Findings indicated that: (1) patients' perceived EE mediated the link between relatives' EE and patients' negative, but not positive, SE; (2) patients' negative SE mediated the effect of patients' perceived EE on positive symptoms and paranoia; (3) the association of relatives' EE with positive symptoms and paranoia was serially mediated by an increased level of patients' perceived EE leading to increases in negative SE; (4) high levels of patients' distress moderated the effect of relatives' EE on symptoms through patients' perceived EE and negative SE. Findings emphasize that patients' SE is relevant for understanding how microsocial environmental factors impact formation and expression of positive symptoms and paranoia in early psychosis. They suggest that broader interventions for patients and their relatives aiming at improving family dynamics might also improve patients' negative SE and symptoms.

## Introduction

Cognitive models of psychosis indicate that low self-esteem (SE) is crucial in the development and persistence of positive symptoms [1–3]. Recent research has demonstrated that low self-

Ethics Committee of the Unió Catalana d'Hospitals (Comitè d'Ètica d'Investigació Clínica (CEIC); number 09-40) and by the Ethics Committee of the Universitat Autònoma de Barcelona (Comissió d'Ètica en l'Experimentació Animal i Humana (CEEAH); number 2679) imposes restrictions for making the data publicly available. Participants agreed for all the data collected to be available to the members of the research group Person-Environment Interaction in Psychopathology led by Prof. Neus Barrantes-Vidal (Address: Departament de Psicologia Clínica i de la Salut, Facultat de Psicologia, Edifici B, Universitat Autònoma de Barcelona, 08193 Cerdanyola del Vallès, Spain; telephone: +34 93 5813864; email: neus.barrantes@uab.cat). Data available on request. Requests should be addressed to the contact details provided above or to the Ethics Committee of the Universitat Autònoma de Barcelona (Comissió d'Ètica en l'Experimentació Animal i Humana, Address: Plaça Acadèmica, Rectorat, Edifici A, Universitat Autònoma de Barcelona, 08193 Cerdanyola del Vallès, Spain; telephone: +34 93 5813578; email: oh.ceea@uab.cat).

**Funding:** Authors are supported by the Spanish Ministerio de Economía y Competitividad (PSI2017-87512-C2-01) and the Comissionat per a Universitats i Recerca de Generalitat de Catalunya (2017SGR1612). N. Barrantes-Vidal is supported by the Institució Catalana de Recerca i Estudis Avançats (ICREA) Academia Award and the Centro de Investigación Biomédica en Red de Salud Mental (CIBERSAM), Instituto de Salud Carlos III, Barcelona, Spain. The funders had no role in study design, data collection and analysis, decision to publish, or preparation of the manuscript.

**Competing interests:** The authors have declared that no competing interests exist.

esteem is related to paranoia and positive symptoms across different stages of the psychosis continuum [4–6]. Although cognitive models of psychosis propose a central role for cognitive/emotional processes as proximal factors to the development of positive symptoms, the influence of environmental factors on the origins and maintenance of symptoms is also postulated. Specifically, Garety et al. [2] indicated that negative or unsupportive family environments might contribute to the development of negative self-beliefs, which in turn may negatively impact patients' clinical outcomes.

Expressed emotion (EE) in psychiatry [7] is a measure of family emotional climate used to describe relatives' attitudes towards a family member with a mental disorder. The presence of high-EE attitudes [i.e., criticism and emotional over-involvement (EOI)] in families is related with poorer clinical outcome in chronic [8–10], first-episode of psychosis [11], and at-risk for psychosis patients [12]. However, there is still debate about the mechanisms linking relatives' EE and patients' symptoms. One of the most supported hypotheses is that patients' high arousal states (e.g., anxiety and/or depression) may act as common pathway mediating the effects of environmental stress (e.g., EE) upon psychotic vulnerability to increase risk of symptoms [e.g., 13] and exacerbate existing symptoms [14–16]. Moreover, following Garety et al. [2], empirical studies have also highlighted the role of patients' self-esteem as a psychological mechanism by which family negative attitudes impact psychotic symptom expression.

Barrowclough et al. [17] showed that the impact of relatives' criticism on schizophrenia patients' negative SE was mediated by its association with patients' reports of negative evaluation by relatives. Furthermore, the association between relatives' criticism and patients' positive symptoms was mediated by its impact on patients' negative SE. In light of these findings, a recent cognitive model of paranoid delusions proposed by Kesting and Lincoln [4] explicitly incorporated the potential influence of negative family environment on patients' self-esteem and ultimately on the origins and course of paranoia. Thus, they conceptualized that self-esteem has a mediating role in the link between adverse interpersonal experiences and paranoid delusions. Empirical studies have supported the expansion of cognitive models of positive symptoms to embrace interpersonal components [e.g., 18] and pointed to the mediating role of SE in the link between negative family environment and patients' symptoms [e.g., 19]. Nevertheless, the relationship between family negative attitudes (i.e., EE) and patients' SE in the prodrome and early psychosis has not been explored. Similarly, no previous early psychosis studies have directly considered the possible mediating role of patients' SE dimensions in the link between EE attitudes and psychotic symptoms. Given that early psychosis is probably the stage when these mechanisms would have a crucial role in exacerbating symptom onset and/or maintenance, the present study aimed to address this important gap in the literature.

Likewise, family positive attitudes (e.g., warmth, positive comments) are related with patients' symptomatic/functional improvement [20–22], and also with higher levels of positive self-evaluation and SE [17,23]. However, no previous studies have investigated the possible contribution of family positive attitudes (e.g., warmth) on patients' SE, and ultimately on patients' clinical outcome, either in chronic or incipient psychosis.

Exploring the putative impact of the interplay between family environment and SE on the development of positive symptoms and paranoia in the early stages of psychosis should provide clearer information than that obtained at more developed stages of the illness, by avoiding many of the confounding effects characteristic of chronic psychosis [24]. Thus, using and comparing at-risk mental state (ARMS) and first-episode psychosis (FEP) participants should improve our ability to distinguish etiologically relevant onset mechanisms from consequences of psychotic disorders. ARMS individuals are predominately characterized by being young help-seeking individuals who experience attenuated positive psychotic symptoms that not reach threshold levels of psychosis. The transition risk to full-blown psychosis is around 22%

at 3 years [25]; being severity of attenuated positive and negative symptoms as well as low functioning the most relevant factors associated with an increased risk [26].

The first goal of the present study was to explore in a sample of patients with ARMS and FEP and their respective relatives whether patients' SE dimensions (positive and negative SE) mediated the effect of relatives' EE dimensions (criticism and EOI) on patients' symptoms (positive symptoms and paranoia) (Fig 1A). We predicted that patients' negative SE would mediate the association between relatives' EE dimensions and patients' symptoms. In the second goal, we tested the Barrowclough's model [17] in an early psychosis sample (patients with ARMS and FEP) by investigating the mediating role of patients' perceived EE (perceived

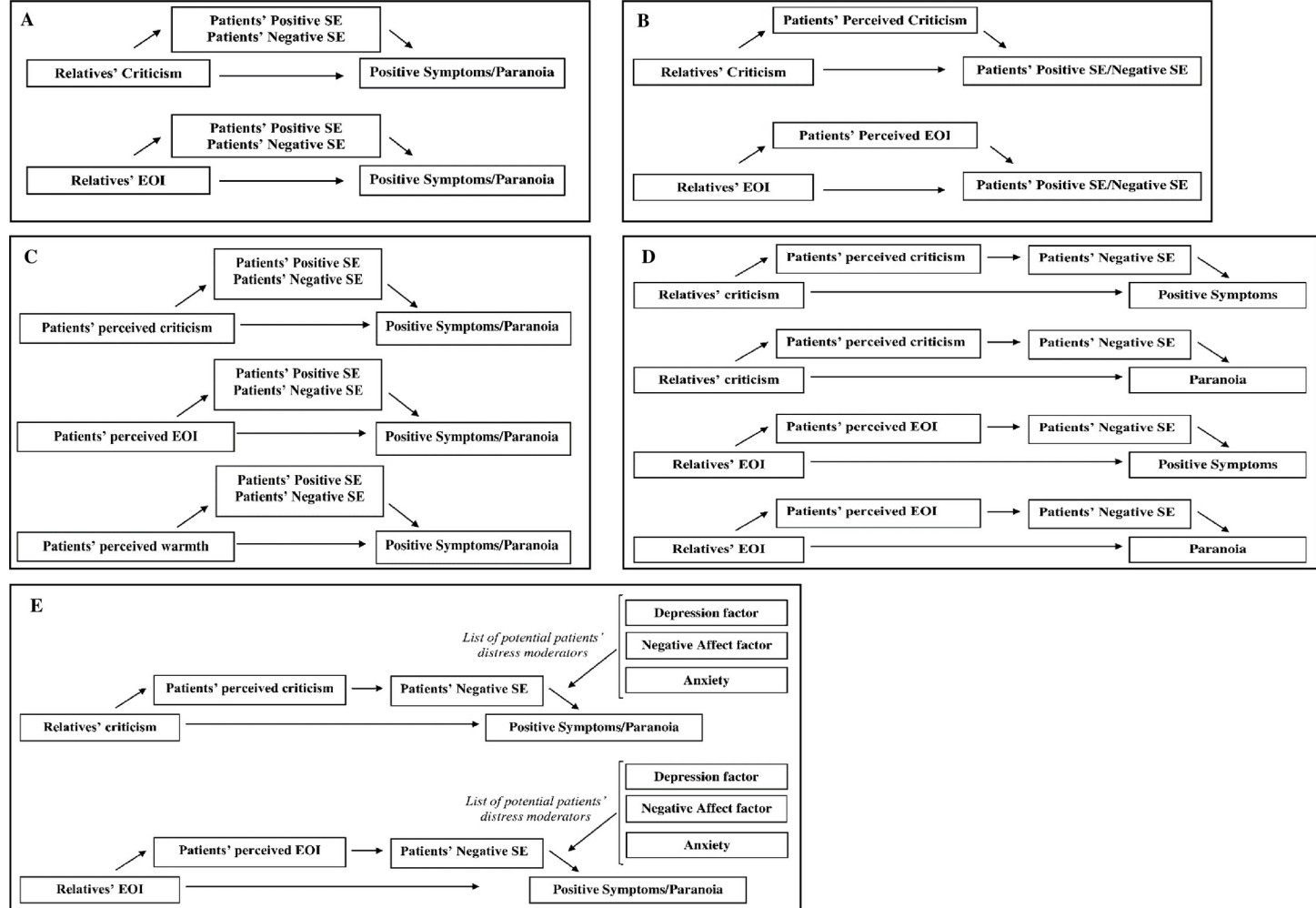

**Fig 1. Conceptual mediation models.** (A) Hypothesized indirect effect of relatives' EE on patients' symptoms via patients' SE. Conceptual multiple mediation model in which is observed the hypothesized indirect effect of relatives' EE dimensions on patients' positive symptoms and paranoia via patients' SE dimensions. (B) Hypothesized indirect effect of relatives' EE on patients' SE via patients' perceived EE. Conceptual simple mediation model in which is observed the hypothesized indirect effect of relatives' EE dimensions on patients' SE dimensions via patients' perceived EE (perceived criticism and perceived EOI). (C) Hypothesized indirect effect of patients' perceived EE on patients' symptoms via patients' SE. Conceptual multiple mediation model in which is observed the hypothesized indirect effect of patients' perceived EE on patients' positive symptoms and paranoia via patients' SE dimensions. (D) Hypothesized indirect effect of relatives' EE on patients' symptoms via patients' perceived EE and patients' SE. Conceptual serial mediation model in which is observed the hypothesized indirect effects of relatives' EE on patients' positive symptoms and paranoia via patients' perceived EE and patients' self-esteem (SE) dimensions. (E) The moderating effect of patients' distress. Conceptual moderated serial mediation model in which the indirect effect of relatives' EE on patients' positive symptoms and paranoia via patients' perceived EE and patients' self-esteem (SE) dimensions is moderated by patients' distress variables (at M2 to Y path).

criticism and EOI) between relatives' reports of EE and patients' SE dimensions (Fig 1B). It was expected that patients' perceived EE would mediate the impact of relatives' EE on patients' negative SE. As patients' perceptions of their relatives' EE have been suggested to be more powerful predictors of outcome than relatives' EE ratings [e.g., 27–29], the third goal was to test the mediating effect of patients' SE dimensions between patients' perceived EE (perceived criticism, EOI and warmth) and symptoms (Fig 1C). We hypothesized that patients' negative SE would mediate the relationship between patients' perceived criticism and EOI with symptoms. Conversely, patients' positive SE was expected to mediate the inverse association between patients' perceived warmth and symptoms. In the fourth goal, a comprehensive model tested the mediating role of patients' perceived EE and patients' negative SE (in a serial causal order) in the link between relatives' EE dimensions and patients' symptoms (Fig 1D). Finally, as high distress states have been also suggested as a mechanism by which relatives' EE impacts on symptom exacerbation, the fifth goal investigated whether patients' distress moderated the effect of relatives' EE on symptoms through patients' perceived EE and negative SE (Fig 1E). In addition, we explored whether these models differed across ARMS and FEP stages, as some mechanisms might be more evident and/or relevant in the at-risk or onset psychosis states.

## Materials and methods

### Participants and procedure

The present study is embedded in a larger longitudinal study carried out in four Mental Health Centers of Barcelona (Spain) conducting the Sant Pere Claver- Early Psychosis Program [30]. Early psychosis patients (ARMS and FEP participants) and their respective relatives were included. ARMS criteria were established based on the Comprehensive Assessment of At-Risk Mental States (CAARMS) [31]. The CAARMS identifies 3 different at-risk mental states groups: the vulnerability group, the APS group, and the BLIPS group. The vulnerability group identifies those individuals with a combination of a trait risk factor and a significant deterioration in social and occupational functioning. The APS group includes those individuals with attenuated psychotic symptoms that not reach threshold levels of psychosis. Finally, the BLIPS group identifies individuals with brief limited intermittent psychotic symptoms that resolved spontaneously without antipsychotic medication. All the ARMS patients were help-seeking individuals, but none of the ARMS patients met DSM-IV-TR criteria [32] for any psychotic disorder or affective disorder with psychotic symptoms. FEP patients met DSM-IV-TR criteria [32] for any psychotic disorder or affective disorder with psychotic symptoms and presented a first-episode of psychosis within the past two years. Mean duration of illness was 11 months (SD = 8.3), 12 months (SD = 8.1) and 12 months (SD = 7.4) for Sample 1, Sample 2 and Sample 3, respectively. However, 2 patients reached a length of 29 months in Sample 1 and 1 patient reached a length of 29 months in Samples 2 and 3. Patient's inclusion criteria were age between 14 and 40 years old and IQ $\geq$ 75. Exclusion criteria for patients were evidence of organically based psychosis and any previous psychotic episode that involved pharmacotherapy. Relatives were referred to the study by their respective affected family members (i.e., early psychosis patients). Patients were informed of the relatives' study and asked to name the person to whom they have a significant/close relationship. After getting the consent of the patient, the relative was contacted and was asked to participate into the study. Thus, the relatives recruited were those who had most regular contact and/or the most significant relationship with the patient. All participants provided written informed consent to participate and completed the assessment protocol within a maximum of 4 weeks. Written informed consent was also obtained from parents of the minors included in the study. The project was developed in

accordance with the Code of Ethics of the World Medical Association (Declaration of Hel-sinki). Ethical approval was granted by the Ethics Committee of the Unió Catalana d'Hospitals (ref. 09–40) and by the Ethics Committee of the Universitat Autònoma de Barcelona (ref. 2679). All the interviews were conducted by experienced clinical psychologists. The time gap between patients and relatives' assessments was minimal (range of 3 to 15 days).

### Measures

Relatives' EE was measured with the Family Questionnaire (FQ) [33], which consists of 20 items equally distributed into two subscales (EOI and criticism). Patients' perceptions of their relatives' EE were measured with the Brief Dyadic Scale of Expressed Emotion (BDSEE) [34], which has three subscales: 'perceived criticism', 'perceived EOI', and 'perceived warmth'.

Self-esteem was assessed with the Rosenberg Self-Esteem Scale (RSES) [35], which has five positively worded items and five negatively worded items. A high total score is indicative of high global self-esteem. Consistent with recent recommendations [17,36,37], we used positive and negative SE dimensional scores. To that end, a principal component analysis (Promax rotation) of the RSES in each the ARMS, FEP, and relatives' samples was conducted (i.e., Sample 1: n = 77; Sample 2: n = 58; Sample 3: n = 93). It revealed a two-factor solution with a large inverse correlation [(Sample 1: r = -0.58); (Sample 2: r = -0.55); (Sample 3: r = -0.60)]. One factor represented positive self-esteem and the other negative self-esteem, explaining 48.24% and 36.09% (Sample 1), 48.92% and 34.13% (Sample 2), 45.84% and 37.53% (Sample 3) of the variance, respectively. Factor scores coefficients were computed for positive and negative trait self-esteem.

Patients' positive symptoms were assessed with the positive subscale of the Positive and Negative Syndrome Scale (PANSS) [38] including the following items: "delusions" (item P1), "conceptual disorganization" (item P2), "hallucinatory behavior" (item P3), "excitement" (item P4), "grandiosity" (item P5), "suspiciousness/persecution" (item P6) and "hostility" (item P7). Paranoia was measured with the "suspiciousness/persecution" item from the PANSS (item P6).

Patient distress was examined from three different perspectives, given that there have been theoretical claims regarding differential contributions of various types of negative emotions [39,40] "pure" general depression, "pure" anxiety, and a mixture of negative affect states. A general measure of depression was derived from a principal component analysis (PCA) of the following measures: Beck Depression Inventory (BDI) [41], Calgary Depression Scale (CDS) [42] and the "depression" item from the PANSS (item G6). S1 Table shows the correlations among these measures and the description of the PCA can be found below it. A general measure of negative affect (NA) was derived from a PCA of the following scales: BDI [41], CDS [42], and the "Depression/Anxiety" factor from the PANSS- Five Factors [43], which encompasses a wide variety of affective-related symptoms. S2 Table displays correlations among these measures and the description of the PCA can be found below it. Anxiety was measured with the "anxiety" item from the PANSS (item G2).

### Statistical analysis

Mediation analyses were performed using PROCESS v2.16 [44]. *Parallel multiple* mediation analyses (model 4; model as a parameter in the PROCESS function) were conducted to examine: (1) the indirect effect of relatives' EE dimensions on symptoms via patients' SE dimensions (goal 1); (2) the indirect effect of patients' perceived EE on symptoms via patients' SE dimensions (goal 3). For each model, the two SE dimensions were entered simultaneously as mediators. *Simple mediation* analyses (model 4) were conducted to examine the indirect effect of

relatives' EE dimensions on patients' SE dimensions via patients' perceived EE (goal 2). *Moderated mediation* analyses (model 59) were performed to explore the indirect effects referred above across ARMS and FEP groups. Moreover, *serial mediation* analyses (model 6) were used to examine the indirect effect of relatives' EE dimensions on symptoms via, first, patients' perceived EE (perceived criticism and EOI), and second, patients' negative SE (goal 4). In the serial analysis, mediators are assumed to have a direct effect on each other [44], and the independent variable (relatives' EE dimensions) is assumed to influence mediators (patients' perceived EE and patients' negative SE) in a serial way that ultimately influences the dependent variable (patients' symptoms). Four different serial mediation models (SMM) were explored: [(SMM$_1$: Relatives' Criticism➡ Perceived Criticism➡ Negative SE➡ Positive Symptoms); (SMM$_2$: Relatives Criticism➡ Perceived Criticism➡ Negative SE➡ Paranoia); (SMM$_3$: Relatives' EOI➡ Perceived EOI➡ Negative SE➡ Positive Symptoms); (SMM$_4$: Relatives EOI➡ Perceived EOI➡ Negative SE➡ Paranoia)]. Finally, PROCESS v3.3 [45] was used to perform the *moderated serial mediation* analyses. These analyses investigated whether the serial mediated model described above was moderated by: (1) patients' distress variables (goal 5) and (2) diagnostic group category (ARMS/FEP) (Model 92).

The moderating role of patients' distress variables in the serial mediation models (SMM) was examined by model 87 which explored the effect of the moderator from path $M_2$ (patients' negative SE) to Y (symptoms). Fig 1 illustrates all the models referred above. The 95% bias-corrected confidence intervals were generated using bootstrapping with 10,000 resamples. Indirect effects were considered significant when the 95% bias-corrected confidence intervals did not include zero.

## Results

Patients and relatives' socio-demographic characteristics as well as descriptive data for all patients and relatives' measures are presented in S3 and S4 Tables, respectively. Depending on study goals, the samples examined differed (e.g., only patients or patient-relative dyads); therefore, there are different numbers of participants in the analyses. For goals examining patient-relative dyads (1, 2, 4 and 5), it was required that the examined measures had been responded by both members of the dyad. Hence, goal 1 included 77 patients (50 ARMS and 27 FEP; Sample 1) and their respective relatives, whereas Goals 2, 4 and 5 included 58 patients (37 ARMS and 21 FEP; Sample 2) and their respective relatives. For goals examining only patients (i.e., goal 3), 93 early psychosis patients (60 ARMS and 33 FEP; Sample 3) were included. Please note that a detailed participant flowchart is available in S1 Fig. As depicted in S1 Fig, there was a total sample of 122 relatives (n = 92 key relatives and n = 30 second closest relatives) of early psychosis patients included at baseline. Taking into account that the present study focused on examining patient-relative dyads, only key relatives (n = 92) were eligible as potential subjects of study. In samples of patient-relative dyads, relatives were mainly female [77.9% (Sample 1); 82.8% (Sample 2)], particularly patient's mothers [75.3% (Sample 1); 81% (Sample 2)]. Mean age of the relatives was 50.71 years old (S.D = 10.8) and 50.69 years old (SD = 11.3) in Sample 1 and 2, respectively. Patients were predominantly male in all samples [70.1% (Sample 1); 72.4% (Sample 2); 68.8% (Sample 3)]. The mean age of the patients was 21.96 years old (S.D = 4.6), 22.05 years old (SD = 4.6) and 22.28 years old (SD = 4.4) in Samples 1, 2 and 3, respectively (please see S3 Table for details about relatives' and patients' socio-demographic characteristics).

### Indirect effects of relatives' EE on symptoms via SE

Pearson's correlations of relatives' EE and symptoms, and of patients' SE with relatives' EE and patients' symptoms are presented in S5 and S6 Tables, respectively. Table 1 displays the results

**Table 1. Mediation analyses examining the indirect effects of relatives' EE on symptoms via positive and negative SE (Sample 1; n = 77).**

| | *IV* = Relatives' Criticism | | | | *IV* = Relatives' EOI | | | |
|---|---|---|---|---|---|---|---|---|
| | | | 95% Bias-corrected CI | | | | 95% Bias-corrected CI | |
| | Raw Parameter Estimate | SE | Lower | Upper | Raw Parameter Estimate | SE | Lower | Upper |
| **Positive symptoms (PANSS)** | | | | | | | | |
| Total Effect | 0.065 | 0.060 | -0.056 | 0.185 | 0.022 | 0.068 | -0.113 | 0.156 |
| Direct Effect | 0.064 | 0.063 | -0.061 | 0.189 | 0.025 | 0.067 | -0.110 | 0.159 |
| Total Indirect Effect | 0.000 | 0.025 | -0.050 | 0.050 | -0.003 | 0.017 | -0.045 | 0.025 |
| Indirect Effect via Positive SE | -0.013 | 0.024 | -0.075 | 0.025 | -0.002 | 0.011 | -0.042 | 0.011 |
| Indirect Effect via Negative SE | 0.013 | 0.017 | -0.007 | 0.071 | -0.001 | 0.015 | -0.039 | 0.025 |
| **Paranoia (PANSS)** | | | | | | | | |
| Total Effect | 0.006 | 0.021 | -0.034 | 0.047 | -0.009 | 0.023 | -0.055 | 0.036 |
| Direct Effect | 0.000 | 0.021 | -0.041 | 0.042 | -0.009 | 0.022 | -0.054 | 0.035 |
| Total Indirect Effect | 0.006 | 0.009 | -0.009 | 0.026 | -0.000 | 0.007 | -0.015 | 0.013 |
| Indirect Effect via Positive SE | 0.001 | 0.008 | -0.014 | 0.019 | 0.000 | 0.004 | -0.006 | 0.011 |
| Indirect Effect via Negative SE | 0.005 | 0.006 | -0.003 | 0.026 | -0.000 | 0.006 | -0.016 | 0.010 |

Note: Results are based on 10,000 bias-corrected bootstrap samples.

of the parallel multiple mediation analyses using relatives' criticism and EOI as independent variables. Two models were tested (one for positive symptoms and one for paranoia) for each of the multiple mediator models. Contrary to our hypotheses, the indirect effect of relatives' criticism on positive symptoms and/or paranoia via SE dimensions was not significant. Likewise, there were no significant indirect effects of relatives' EOI via SE dimensions on positive symptoms and/or paranoia. Moderated mediation analyses revealed that group (ARMS vs. FEP) did not moderate any of these effects (S7 Table).

## Indirect effects of relatives' EE on SE via perceived EE

The correlation coefficients between relatives' EE and SE as well as of perceived EE with relatives' EE and symptoms are in S8 and S9 Tables, respectively. The first simple mediation models tested how relatives' criticism was related with SE dimensions via its effect on perceived criticism. The second mediation analyses examined whether relatives' EOI was related with SE dimensions via its effect on perceived EOI (Table 2). Two models were tested (for negative SE and for positive SE) for each mediator model. As expected, there was a significant indirect effect of relatives' criticism on negative SE (but not on positive SE) via perceived criticism. Likewise, relatives' EOI was related with negative, but not positive, SE indirectly through perceived EOI. However, the direct effect of relatives' criticism and relatives' EOI on negative SE (controlling for the mediator) was nonsignificant.

Results of the moderated mediation analyses examining the effect of group revealed that the effect of relatives' criticism on negative SE was mediated by perceived criticism in FEP (conditional IE = 0.0293, SE = 0.0207, LLCI = 0.0011, ULCI = 0.0852) but not in ARMS participants (conditional IE = 0.0171, SE = 0.0123, LLCI = -0.0305, ULCI = 0.0625). However, the conditional IE was not significantly different across the two groups [Index of moderated mediation (IMM) = 0.0122, SE = 0.0241, LLCI = -0.0305, ULCI = 0.0625]. Conversely, the effect of relatives' EOI on negative SE was mediated by perceived EOI in ARMS (conditional IE = 0.0347, SE = 0.0165, LLCI = 0.0092, ULCI = 0.0749) but not in FEP individuals (conditional IE = 0.0035, SE = 0.0242, LLCI = -0.0494, ULCI = 0.0469). Nevertheless, the conditional IE

**Table 2. Mediation analyses examining the indirect effects of relatives' EE on SE via perceived EE (Sample 2; n = 58).**

| | Raw Parameter Estimate | SE | 95% Bias-corrected Confidence Interval | |
| --- | --- | --- | --- | --- |
| | | | Lower | Upper |
| *IV = Relatives' Criticism* | | | | |
| **Negative SE (RSES)** | | | | |
| Total Effect | 0.026 | 0.021 | -0.016 | 0.069 |
| Direct Effect | 0.005 | 0.022 | -0.040 | 0.049 |
| Indirect Effect via Perceived Criticism | **0.022**[*] | 0.009 | 0.006 | 0.045 |
| **Positive SE (RSES)** | | | | |
| Total Effect | -0.039 | 0.021 | -0.081 | 0.003 |
| Direct Effect | -0.032 | 0.023 | -0.078 | 0.014 |
| Indirect Effect via Perceived Criticism | -0.007 | 0.011 | -0,028 | 0.014 |
| *IV = Relatives' EOI* | | | | |
| **Negative SE (RSES)** | | | | |
| Total Effect | -0.001 | 0.023 | -0.046 | 0.045 |
| Direct Effect | -0.024 | 0.021 | -0.066 | 0.018 |
| Indirect Effect via Perceived EOI | **0.024**[*] | 0.012 | 0.003 | 0.052 |
| **Positive SE (RSES)** | | | | |
| Total Effect | -0.005 | 0.023 | -0.050 | 0.040 |
| Direct Effect | 0.009 | 0.023 | -0.036 | 0.055 |
| Indirect Effect via Perceived EOI | -0.015 | 0.013 | -0.048 | -0.000 |

Note: Results are based on 10,000 bias-corrected bootstrap samples.

[*] 95% Confidence Interval does not include zero.

was not significantly different across the two groups (IMM = -0.0311, SE = 0.0294, LLCI = -0.0946, ULCI = 0.0196).

## Indirect effects of perceived EE on symptoms via SE

Pearson's correlations between perceived EE and symptoms as well as among SE, perceived EE and symptoms are in S10 and S11 Tables, respectively. The parallel multiple mediation analyses using perceived criticism, EOI, and warmth as independent variables are in Table 3. Two models were tested (for positive symptoms and paranoia) for each of the multiple mediator models. Results revealed that there was a significant indirect effect of perceived criticism as well as of perceived EOI on both positive symptoms and paranoia via negative SE, but not via positive SE, as expected. Note that the direct relationship of perceived criticism with positive symptoms and paranoia no longer remained significant when the indirect pathway through patients' negative SE was included. Finally, in contrast to our hypotheses, the indirect effect of perceived warmth on positive symptoms and/or paranoia via SE dimensions was not significant.

Moderated mediation analyses revealed that the effect of perceived criticism on positive symptoms was mediated by both SE dimensions (negative SE and positive SE) in FEP patients, but not in ARMS patients (Table 4). The magnitude of the conditional IE differed significantly between the two groups, indicating that the indirect effect of perceived criticism on positive symptoms through both SE dimensions was significantly different between ARMS and FEP patients. Similarly, the effect of perceived criticism on paranoia was mediated by negative SE in FEP patients, but not in ARMS. The magnitude of the conditional IE was significantly different across the two groups. Group did not moderate the indirect effect of perceived EOI on

**Table 3. Mediation analyses examining the indirect effects of perceived EE on symptoms via positive and negative SE (Sample 3; n = 93).**

| | *IV* = Perceived criticism | | | | *IV* = Perceived EOI | | | | *IV* = Perceived warmth | | | |
|---|---|---|---|---|---|---|---|---|---|---|---|---|
| | | | 95% Bias-corrected CI | | | | 95% Bias-corrected CI | | | | 95% Bias-corrected CI | |
| | Raw Parameter Estimate | SE | Lower | Upper | Raw Parameter Estimate | SE | Lower | Upper | Raw Parameter Estimate | SE | Lower | Upper |
| **Positive symptoms (PANSS)** | | | | | | | | | | | | |
| Total Effect | **0.076***  | 0.038 | 0.001 | 0.150 | 0.049 | 0.028 | -0.007 | 0.106 | -0.032 | 0.042 | -0.115 | 0.051 |
| Direct Effect | 0.063 | 0.039 | -0.013 | 0.140 | 0.037 | 0.029 | -0.021 | 0.095 | -0.022 | 0.042 | -0.106 | 0.061 |
| Total Indirect Effect | 0.012 | 0.013 | -0.009 | 0.044 | 0.013 | 0.011 | -0.005 | 0.038 | -0.009 | 0.014 | -0.043 | 0.013 |
| Indirect Effect via Positive SE | -0.011 | 0.013 | -0.049 | 0.005 | -0.008 | 0.010 | -0.039 | 0.005 | 0.011 | 0.014 | -0.007 | 0.052 |
| Indirect Effect via Negative SE | **0.023***  | 0.015 | 0.001 | 0.065 | **0.020***  | 0.013 | 0.001 | 0.052 | -0.020 | 0.018 | -0.070 | 0.001 |
| **Paranoia (PANSS)** | | | | | | | | | | | | |
| Total Effect | **0.028***  | 0.012 | 0.005 | 0.052 | 0.006 | 0.009 | -0.012 | 0.025 | **-0.029***  | 0.013 | -0.056 | -0.004 |
| Direct Effect | 0.020 | 0.012 | -0.004 | 0.044 | -0.002 | 0.009 | -0.021 | 0.016 | -0.024 | 0.013 | -0.049 | 0.002 |
| Total Indirect Effect | **0.008***  | 0.005 | 0.000 | 0.021 | 0.009 | 0.004 | 0.003 | 0.019 | -0.006 | 0.005 | -0.019 | 0.003 |
| Indirect Effect via Positive SE | -0.001 | 0.004 | -0.012 | 0.005 | -0.001 | 0.003 | -0.008 | 0.005 | 0.002 | 0.005 | -0.005 | 0.014 |
| Indirect Effect via Negative SE | **0.010***  | 0.006 | 0.001 | 0.026 | **0.009***  | 0.005 | 0.002 | 0.022 | -0.008 | 0.006 | -0.025 | 0.001 |

Note: Results are based on 10,000 bias-corrected bootstrap samples.

*95% Confidence Interval does not include zero-.

positive symptoms via SE. However, the effect of perceived EOI on paranoia was mediated by negative SE in FEP but not in ARMS patients. In this case, the conditional IE did not differ between ARMS and FEP patients.

As mentioned previously, no significant indirect effects were observed for the model testing the effects of perceived warmth on positive symptoms and/or paranoia via SE dimensions. However, when the effect of the moderator was examined, results revealed that the effect of perceived warmth on both positive symptoms and paranoia was mediated by positive SE in FEP but not in ARMS, and the magnitude of the conditional IE was significantly different across the two groups.

### Serial Mediation Models (SMM)

As illustrated in Fig 1D, a series of serial multiple mediation models were explored using perceived EE factors and negative SE as mediators. In serial mediation, mediators are assumed to have a direct effect on each other [44], and the independent variable (relatives' criticism/EOI) is assumed to influence mediators in a serial way that ultimately influences the dependent variable. Results revealed four significant indirect pathways (Table 5). First, there were two significant indirect pathways from relatives' criticism to positive symptoms (SMM1) and paranoia (SMM2) through perceived criticism and negative SE. This means that increased relatives' criticism increases perceived criticism that in turn increases negative SE and results in increased positive symptoms and paranoia. Second, there were two significant indirect pathways from relatives' EOI to positive symptoms (SMM3) and paranoia (SMM4) through perceived EOI and negative SE. Thus, relatives' EOI was serially associated to perceived EOI and negative SE,

**Table 4. Conditional indirect effects of perceived EE on symptoms through positive and negative SE (Sample 3; n = 93).**

| | | | | | | | | | | | | |
|---|---|---|---|---|---|---|---|---|---|---|---|---|
| | | | | | **Conditional indirect effects at different values of the moderator** | | | | **Index of moderated mediation** | | | |
| | | | | | | | **95% Bias-corrected CI** | | | | **95% Bias-corrected CI** | |
| **Predictor** | **Outcome** | **Mediator** | **Moderator** | **Level** | **Raw Parameter Estimate** | **SE** | **Lower** | **Upper** | **Index** | **SE** | **Lower** | **Upper** |
| Perceived Criticism | Positive symptoms | Positive SE | Group | ARMS | 0.001 | 0.010 | -0.013 | 0.032 | **-0.088***  | **0.072** | **-0.289** | **-0.001** |
| | | | | FEP | **-0.087*** | **0.071** | **-0.291** | **-0.002** | | | | |
| | | Negative SE | Group | ARMS | 0.004 | 0.011 | -0.007 | 0.048 | **0.076*** | **0.047** | **0.006** | **0.208** |
| | | | | FEP | **0.080*** | **0.045** | **0.014** | **0.208** | | | | |
| Perceived Criticism | Paranoia | Positive SE | Group | ARMS | 0.002 | 0.005 | -0.003 | 0.020 | -0.026 | 0.024 | -0.089 | 0.004 |
| | | | | FEP | -0.024 | 0.024 | -0.088 | 0.004 | | | | |
| | | Negative SE | Group | ARMS | 0.002 | 0.004 | -0.003 | 0.018 | **0.034*** | **0.018** | **0.007** | **0.081** |
| | | | | FEP | **0.036*** | **0.017** | **0.010** | **0.081** | | | | |
| Perceived EOI | Positive symptoms | Positive SE | Group | ARMS | 0.002 | 0.011 | -0.014 | 0.035 | -0.035 | 0.037 | -0.128 | 0.008 |
| | | | | FEP | -0.033 | 0.035 | -0.128 | 0.001 | | | | |
| | | Negative SE | Group | ARMS | 0.004 | 0.013 | -0.016 | 0.040 | 0.035 | 0.035 | -0.015 | 0.125 |
| | | | | FEP | 0.039 | 0.033 | -0.000 | 0.129 | | | | |
| Perceived EOI | Paranoia | Positive SE | Group | ARMS | 0.003 | 0.005 | -0.002 | 0.019 | -0.013 | 0.013 | -0.044 | 0.002 |
| | | | | FEP | -0.010 | 0.012 | -0.042 | 0.001 | | | | |
| | | Negative SE | Group | ARMS | 0.003 | 0.006 | -0.005 | 0.019 | 0.014 | 0.013 | -0.008 | 0.043 |
| | | | | FEP | **0.017*** | **0.012** | **0.001** | **0.046** | | | | |
| Perceived Warmth | Positive symptoms | Positive SE | Group | ARMS | -0.001 | 0.016 | -0.051 | 0.020 | **0.081*** | **0.066** | **0.000** | **0.278** |
| | | | | FEP | **0.079*** | **0.064** | **0.006** | **0.287** | | | | |
| | | Negative SE | Group | ARMS | -0.006 | 0.016 | -0.067 | 0.009 | -0.049 | 0.051 | -0.170 | 0.031 |
| | | | | FEP | -0.056 | 0.048 | -0.174 | 0.013 | | | | |
| Perceived Warmth | Paranoia | Positive SE | Group | ARMS | -0.002 | 0.006 | -0.027 | 0.004 | **0.030*** | **0.024** | **0.001** | **0.101** |
| | | | | FEP | **0.027*** | **0.023** | **0.001** | **0.099** | | | | |
| | | Negative SE | Group | ARMS | -0.002 | 0.006 | -0.023 | 0.004 | -0.021 | 0.018 | -0.061 | 0.013 |
| | | | | FEP | -0.023 | 0.018 | -0.064 | 0.007 | | | | |

Note: Results are based on 10,000 bias-corrected bootstrap samples.

[a]Perceived Criticism, Perceived EOI, Perceived Warmth (X-Independent variable) and Diagnostic Category (W-moderator) were mean centered prior to analyses.

*95% Confidence Interval does not include zero.

resulting in increased positive symptoms and paranoia. Moderated serial mediation analyses indicated that group (ARMS, FEP) did not moderate any of these effects (S12 Table).

**Table 5. Indirect effects for the paths on the Serial Mediation Models (SMMs) (Sample 2; n = 58).**

| | | | **95% Bias-corrected Confidence Interval** | |
|---|---|---|---|---|
| | **Raw Parameter Estimate** | **SE** | **Lower** | **Upper** |
| **SMM1:** Relatives' Criticism➔ Perceived Criticism➔ Negative SE➔ Positive Symptoms | **0.019*** | 0.014 | 0.002 | 0.062 |
| **SMM2:** Relatives Criticism➔ Perceived Criticism➔ Negative SE➔ Paranoia | **0.008*** | 0.005 | 0.001 | 0.025 |
| **SMM3:** Relatives' EOI➔ Perceived EOI➔ Negative SE➔ Positive Symptoms | **0.021*** | 0.015 | 0.001 | 0.067 |
| **SMM4:** Relatives EOI➔ Perceived EOI➔ Negative SE➔ Paranoia | **0.010*** | 0.007 | 0.001 | 0.029 |

Note: Results are based on 10,000 bias-corrected bootstrap samples.

*95% Confidence Interval does not include zero.

## The role of patients' distress in moderating the effect of relatives' EE on symptoms via perceived EE and negative SE

As shown in Table 6, findings indicated that high levels of depressive and NA symptoms (but not anxiety symptoms) moderated: (1) the indirect effect of relatives' criticism on positive symptoms through perceived criticism and negative SE (SMM1), and (2) the indirect effect of relatives' EOI on positive symptoms through perceived EOI and negative SE (SMM3). These results suggested that the effect of relatives' criticism/EOI on positive symptoms via perceived criticism/EOI and negative SE is observed when depressive and NA symptoms are high (1 SD above the mean) but not low (1SD below the mean). Furthermore, the magnitude of the conditional IE (as indicated by the IMM) differed between high and low levels of both depressive and NA symptoms. This provided further evidence that the above-mentioned indirect effects were significantly different between those individuals who had high depressive/NA symptoms and those who had low depressive/NA symptoms.

**Table 6. Conditional indirect effects of relatives' EE on symptoms through perceived EE (M$_1$) and negative SE (M$_2$) (Sample 2; n = 58).**

| | Conditional indirect effects at different values of the moderator | | | | | | Index of moderated mediation | | | |
| | | | | | 95% Bias-corrected CI | | | | 95% Bias-corrected CI | |
| | Moderator | Level | Raw Parameter Estimate | SE | Lower | Upper | Index | SE | Lower | Upper |
|---|---|---|---|---|---|---|---|---|---|---|
| **SMM1: Relatives' Criticism➤ Perceived Criticism➤ Negative SE➤ Positive Symptoms** | Depression | Low | -0.006 | 0.013 | -0.029 | 0.025 | **0.023**\* | **0.013** | **0.002** | **0.053** |
| | | High | **0.039**\* | **0.025** | **0.004** | **0.099** | | | | |
| | Negative Affect | Low | -0.006 | 0.013 | -0.029 | 0.025 | **0.022**\* | **0.013** | **0.002** | **0.052** |
| | | High | **0.038**\* | **0.025** | **0.004** | **0.099** | | | | |
| | Anxiety | Low | 0.011 | 0.018 | -0.024 | 0.048 | 0.006 | 0.008 | -0.008 | 0.025 |
| | | High | 0.024 | 0.017 | -0.002 | 0.064 | | | | |
| **SMM2: Relatives Criticism➤ Perceived Criticism➤ Negative SE➤ Paranoia** | Depression | Low | 0.000 | 0.005 | -0.009 | 0.012 | 0.006 | 0.005 | -0.001 | 0.017 |
| | | High | 0.012 | 0.009 | -0.000 | 0.034 | | | | |
| | Negative Affect | Low | -0000 | 0.005 | -0.009 | 0.011 | 0.006 | 0.005 | -0.001 | 0.018 |
| | | High | 0.012 | 0.009 | -0.000 | 0.035 | | | | |
| | Anxiety | Low | 0.005 | 0.006 | -0.008 | 0.018 | 0.004 | 0.003 | -0.001 | 0.011 |
| | | High | **0.012**\* | **0.008** | **0.001** | **0.030** | | | | |
| **SMM3: Relatives' EOI➤ Perceived EOI➤ Negative SE➤ Positive Symptoms** | Depression | Low | -0.006 | 0.015 | -0.037 | 0.026 | **0.025**\* | **0.016** | **0.000** | **0.060** |
| | | High | **0.043**\* | **0.029** | **0.001** | **0.112** | | | | |
| | Negative Affect | Low | -0.006 | 0.015 | -0.036 | 0.026 | **0.024**\* | **0.015** | **0.000** | **0.059** |
| | | High | **0.042**\* | **0.028** | **0.000** | **0.107** | | | | |
| | Anxiety | Low | 0.014 | 0.021 | -0.029 | 0.054 | 0.005 | 0.010 | -0.013 | 0.031 |
| | | High | 0.025 | 0.020 | -0.005 | 0.073 | | | | |
| **SMM4: Relatives' EOI➤ Perceived EOI➤ Negative SE➤ Paranoia** | Depression | Low | 0.002 | 0.006 | -0.009 | 0.016 | 0.007 | 0.006 | -0.002 | 0.020 |
| | | High | 0.014 | 0.011 | -0.000 | 0.041 | | | | |
| | Negative Affect | Low | 0.001 | 0.006 | -0.009 | 0.015 | 0.007 | 0.006 | -0.001 | 0.021 |
| | | High | 0.015 | 0.011 | -0.000 | 0.042 | | | | |
| | Anxiety | Low | 0.006 | 0.007 | -0.007 | 0.023 | 0.004 | 0.004 | -0.002 | 0.013 |
| | | High | **0.015**\* | **0.010** | **0.000** | **0.039** | | | | |

Note: Results are based on 10,000 bias-corrected bootstrap samples.

[a]Relatives' Criticism, Relatives' EOI (X-Independent variable) and Diagnostic Category (W-moderator) were mean centered prior to analyses.

\*95% Confidence Interval does not include zero.

Conversely, analyses revealed that high levels of anxiety symptoms (but not depressive or NA symptoms) moderated: (1) the indirect effect of relatives' criticism on paranoia through perceived criticism and negative SE (SMM2), and (2) the indirect effect of relatives' EOI on paranoia through perceived EOI and negative SE (SMM4). Hence, the indirect effects of relatives' criticism/EOI on paranoia was observed when anxiety levels are high (1 SD above the mean) but not when anxiety symptoms are low (1SD below the mean). However, the conditional IE (as indicated by the IMM) did not differ across low and high levels of anxiety. This suggested that the above-mentioned indirect effects did not differ significantly across low and high levels of patients' anxiety.

## Discussion

The present study emphasizes the importance of considering the interplay between microsocial environmental factors such as family dynamics and SE in the formation and/or expression of positive symptoms and paranoia in the critical period of the emergence of psychosis. To the best of our knowledge, the effects of relatives' EE and patients' perceived EE on symptomatology via patient' SE have not been previously explored in early psychosis. Parallel mediation analyses provided a sophisticated approach for independently examining the impact of relatives' EE dimensions and perceived EE on symptoms via SE dimensions, indicating that only perceived EE, but not relatives' EE ratings, impacted negatively on positive symptoms and paranoia via negative SE. However, when all these variables were simultaneously analyzed in a comprehensive serial mediation model, our results revealed, for the first time, that relatives' EE ratings were serially associated with perceived EE and negative SE, resulting in increased positive symptoms and paranoia. In addition, the current study provides a novel contribution by indicating that patients' distress moderated the effect of relatives' EE on symptoms through its impact on perceived EE and negative SE. Our findings also revealed that negative, but not positive, SE was the most common mediating factor between EE and symptoms, suggesting that negative SE may be especially related to positive symptoms and paranoia [46], and highlighting the significance of separately exploring positive and negative SE [17,36,37]. Finally, this study emphasizes how the interplay between family environment and SE is related to the expression of symptoms across ARMS and FEP stages, thereby enabling detection of meaningful differences in these mechanisms across risk and first episode phases.

### The effect of relatives' EE and perceived EE

In contrast to our expectations and previous research [17], SE dimensions did not emerge as mediators between relatives' EE and symptoms. However, in accordance with results from Barrowclough et al. [17], relatives' criticism did have an effect on negative SE through subjective appraisals of such family attitudes. Furthermore, the effect of relatives' EOI on negative SE was mediated by the influence of the EOI on the patient. This suggests that the impact of relatives' EE on patients' SE might only occur when critical and/or EOI attitudes from family members are salient to an individual's self-evaluation, suggesting that patients' subjective appraisals of EE are more relevant to their SE than relatives' EE itself.

### The effect of perceived EE on symptoms via SE

Given that perceived EE mediates patients' SE [17] and low SE impacts the formation of symptoms, as previously suggested by both theoretical and empirical research [1–3], then perceived EE may be a better predictor of outcome than relatives' EE itself, and thus a more sensitive predictor of symptom exacerbation [e.g., 27–29].

Consistent with our hypotheses, parallel mediation analyses indicated that perceived criticism had an indirect effect on positive symptoms and paranoia through negative, but not positive, SE. Drawing from previous models [17–19], our findings suggest that continued perceptions of critical attitudes from family members might foster an internalization of criticism (e.g., self-criticism). Such continued self-criticism could trigger beliefs of inferiority about the self (e.g., dysfunctional self-concepts such as *"I am bad"*, *"I am useless"*) and decrease SE, thus rendering individuals more susceptible to mistrusting others' intentions or perceiving the world as dangerous. Dysfunctional beliefs about the self could be projected to interpersonal relationships, thus contributing to the emergence of cognitive and perceptual disturbances (e.g., the self is experienced as bad, leading to ideas that others will criticize me) [e.g., 2].

The effect of perceived EOI on positive symptoms and paranoia via negative SE, suggests that continued perceptions of EOI from family members (e.g., worry, controlling behaviors, continued self-sacrifice) could contribute to the perception of less autonomy and self-governance in the patient (i.e., negative beliefs about the self). These results support previous findings [19] and are consistent with the model that negative beliefs about the self may evolve into negative evaluations of others that may influence paranoid ideation, delusional beliefs or perceptual disturbances [e.g., 47].

## The moderating role of group

Perceived criticism mediated the effect of relatives' criticism on negative SE in FEP patients but not in ARMS patients. It is likely that FEP, unlike ARMS patients, have had a continued exposure to relatives' critical attitudes during both the at risk and FEP stages that might produce deleterious effects on their SE because of the cumulative impact of social stress. Conversely, perceived EOI mediated the effect of relatives' EOI on negative SE in ARMS but not FEP patients. A possible explanation might be that relatives of ARMS patients are exposed for the first time to symptoms and impairment, and this may trigger the onset of EOI attitudes more so in ARMS than FEP relatives. Therefore, ARMS patients may suddenly perceive intrusive and excessively protective attitudes from their caregivers that might threaten their appraisals of individual autonomy and negatively influence their SE. These moderated mediation results deserve their own interpretation because significant conditional IEs were observed for the FEP and ARMS group. However, we cannot reject the null hypothesis that the mentioned IEs differed significantly between the two groups, as the magnitude of the conditional IEs was not significantly different across ARMS and FEP groups.

The effect of perceived criticism on positive symptoms and paranoia was mediated by SE in FEP but not in ARMS patients. As suggested, these differences might relate to a longer exposure to criticism experienced by FEP patients, and the cumulative effect of criticism might provoke a greater negative SE impairment and ultimately a deleterious impact on positive symptoms and paranoia. Importantly, both positive and negative SE mediated the impact of perceived criticism on positive symptoms in FEP patients. It may be that prolonged exposure to critical attitudes could impair both negative and positive beliefs about the self in FEP patients, and also to invoke the notion that repeated exposure to environmental stressors (e.g., critical attitudes) sensitizes the behavioral stress response to subsequent reexposures (i.e., behavioral sensitization), implying an increased psychotic reactivity to stress [48,49]. On the other hand, the effect of perceived EOI on paranoia was mediated by negative SE in FEP but not in ARMS patients. At first instance, this result could seem counterintuitive given that the present study has also shown that relatives' EOI had a stronger negative influence on ARMS patients' negative SE via perceived EOI. Thus, one might expect that perceived EOI also had a more negative impact on symptoms in ARMS, but this is not the case. Therefore, it seems that

although EOI is more detrimental for ARMS patients' SE, this does not yet lead to the worsening their symptoms. Presumably, given that patients with longer-lasting and more severe psychotic symptoms (i.e., FEP patients in comparison to ARMS) tend to be more sensitive to environmental stress -probably because of behavioral sensitization processes- [50,51], it is likely that FEP individuals show increased emotional and psychotic reactivity to family negative attitudes.

Finally, the effect of perceived warmth on both positive symptoms and paranoia was mediated by positive SE in FEP but not in ARMS. These results suggest that FEP patients, characterized by heightened environmental susceptibility [e.g., 52] display enhanced sensitivity to negative family environments and positive family environments, which has therapeutic implications and highlights the relevance of social support as a protective factor of psychosis [53].

## The effect of relatives' EE on symptoms via perceived EE and negative SE

Given that EE reflects a transactional process between patients and relatives [54], patients' perceptions of their relatives' attitudes are as important as relatives' attitudes. Therefore, we tested a multiple serial mediation model encompassing relatives' EE, perceived EE, SE and symptoms. Our results indicated that relatives' criticism/EOI was serially associated to perceived criticism/EOI and negative SE, which resulted in increased positive symptoms and paranoia. This means that relatives' EE attitudes impacts symptoms through the cumulative effect exerted on perceived EE and negative SE.

Note that the effect of relatives' EE on symptoms via SE dimensions was not significant. Thus, patients' subjective appraisals of their family environment (perceived EE) appear fundamental for understanding the association of relatives' EE and patients' outcomes. This challenges some traditional EE research that described the patient as passively experiencing relative's EE attitudes, and highlights the relevance of subjective appraisals. Hence, the current results extend previous research [17,18] by showing that the dyadic view of EE is relevant to enrich our understanding of the mechanisms leading to the impairment of both SE and symptoms.

## The moderating effect of patients' distress

The hypothesized moderated serial mediation models sought to clarify whether the impact of EE attitudes on the subsequent cognitive responses and psychotic symptoms is moderated by a final affective reaction (i.e., patients' distress variables), which in turn could be deemed as a "final trigger" of psychotic symptoms.

Results indicated a separate emotional pathway to overall positive symptoms and paranoia. Specifically, high levels of depressive and NA symptoms (but not anxiety) moderated the effect of relatives' EE on positive symptoms via perceived criticism/EOI and negative SE. The magnitude of the conditional IE was significantly different across high and low levels of both depressive and NA symptoms. Conversely, results showed that high levels of anxiety (but not depression or NA) moderated the effect of relatives' EE on paranoia via perceived criticism/EOI and negative SE, which emphasize the specific role of anxiety in the development of paranoia [40,55]. However, the conditional IE (as indicated by the IMM) did not differ across low and high levels of anxiety.

These findings are line with prominent theories [14,15] that negative emotional states are critical elements influencing the association between a negative family environment and positive symptoms [16]. Moreover, our results are broadly consistent with the postulated affective pathway from negative SE to positive symptoms via negative affect [2,56,57], and also with the combined cognitive and affective pathway to positive symptoms proposed by Garety et al. [2].

In this combined pathway, triggering events (e.g., family negative attitudes) result in the disruption of both cognitive (e.g., negative beliefs about the self) and affective processes (i.e., negative emotional states), which in turn lead to the formation of positive symptoms. Overall, these findings add to a growing research showing that environmental stressors result in negative SE, and that negative SE precedes and triggers symptoms via negative emotional states.

Regarding limitations, the cross-sectional design limits causal inferences, which require longitudinal studies. Due to limitations on the sample size, findings and conclusions from the present study must be interpreted. Also, the use of self-report measures of relatives' EE, perceived EE, SE as well as some variables of patients' distress were assessed using a self-reporting mechanism, additional observed-based rating of these constructs would have allowed for a more differentiated view. Finally, given that the EE construct is conceptualized within an interactional framework [58,59], it is crucial that future studies examine how EE attitudes are related to patients' SE and clinical outcomes in real time as relatives and patients navigate their real-life settings.

The present study provides new insights into the critical microenvironmental factor of family dynamics, as well as psychological mechanisms underlying the early manifestation of positive symptoms and paranoia. Collectively, findings indicated that patients' negative SE is relevant for how family negative attitudes (based on the subjective appraisals of both relatives and patients) impact the formation of positive symptoms and paranoia in the early stages of the disorder. Furthermore, patients' negative emotional states are relevant for understanding these associations and offer promising targets for prophylactic interventions. These findings suggest that broader interventions for patients and their relatives that aim at improving family atmosphere might be able to improve patients' SE and reduce or prevent negative clinical outcomes.

## Supporting information

**S1 Fig. Flow chart describing the participants included in the study.**
(DOCX)

**S1 Table. Pearson correlations among BDI, CDS and PANSS-Depression (n = 58).**
(DOCX)

**S2 Table. Pearson correlations among BDI, CDS and PANSS-5 Factors-Depression/Anxiety Scale (n = 58).**
(DOCX)

**S3 Table. Descriptive data on socio-demographic characteristics of early psychosis patients and their respective relatives.**
(DOCX)

**S4 Table. Descriptive data of early psychosis patients and their respective relatives.**
(DOCX)

**S5 Table. Pearson correlations of relatives' EE with patients' symptoms (n = 77).**
(DOCX)

**S6 Table. Pearson correlations of patients' SE with relatives' EE and patients' symptoms (n = 77).**
(DOCX)

**S7 Table. Conditional indirect effects of relatives' EE on symptoms through positive and negative SE (n = 77).**
(DOCX)

**S8 Table. Pearson correlations of relatives' EE with patients' SE (n = 58).**
(DOCX)

**S9 Table. Pearson correlations of patients' perceived EE with relatives' EE and patients' SE (n = 58).**
(DOCX)

**S10 Table. Pearson correlations of perceived EE with symptoms (n = 93).**
(DOCX)

**S11 Table. Pearson correlations of patients' SE with patients' perceived EE and patients' symptoms (n = 93).**
(DOCX)

**S12 Table. Conditional indirect effects of relatives' EE on symptoms through perceived EE (M1) and negative SE (M2) (n = 58).**
(DOCX)

## Acknowledgments

The authors appreciate the support offered by the clinicians and all members of the staff of the Fundació Sanitària Sant Pere Claver who provided access to the families that participated in the study. Specially thanks to the patients and their respective relatives who consented to participate. We acknowledge Tecelli Domínguez-Martínez, Paula Cristóbal-Narváez and Cristina Medina-Pradas for their participation in the data collection.

## Author Contributions

**Conceptualization:** Lídia Hinojosa-Marqués, Manel Monsonet, Thomas R. Kwapil, Neus Barrantes-Vidal.

**Data curation:** Lídia Hinojosa-Marqués, Manel Monsonet, Thomas R. Kwapil, Neus Barrantes-Vidal.

**Formal analysis:** Lídia Hinojosa-Marqués, Manel Monsonet, Thomas R. Kwapil, Neus Barrantes-Vidal.

**Funding acquisition:** Neus Barrantes-Vidal.

**Investigation:** Lídia Hinojosa-Marqués, Manel Monsonet, Neus Barrantes-Vidal.

**Methodology:** Thomas R. Kwapil, Neus Barrantes-Vidal.

**Project administration:** Neus Barrantes-Vidal.

**Resources:** Neus Barrantes-Vidal.

**Software:** Thomas R. Kwapil, Neus Barrantes-Vidal.

**Supervision:** Thomas R. Kwapil, Neus Barrantes-Vidal.

**Validation:** Thomas R. Kwapil.

**Visualization:** Lídia Hinojosa-Marqués, Manel Monsonet, Neus Barrantes-Vidal.

**Writing – original draft:** Lídia Hinojosa-Marqués, Manel Monsonet.

**Writing – review & editing:** Lídia Hinojosa-Marqués, Manel Monsonet, Thomas R. Kwapil, Neus Barrantes-Vidal.

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
