## [Decision Letter · Decision Letter 0]

14 Dec 2020

PONE-D-20-31530

The impact of family environment on self-esteem and symptoms in early psychosis

PLOS ONE

Dear Dr. Barrantes-Vidal,

Thank you for submitting your manuscript to PLOS ONE. After careful consideration, we feel that it has merit but does not fully meet PLOS ONE’s publication criteria as it currently stands. Therefore, we invite you to submit a revised version of the manuscript that addresses the points raised during the review process. Both reviewers think your paper is very interesting but could be improved further by addressing some methodological issues.

We look forward to receiving your revised manuscript.

Kind regards,

Therese van Amelsvoort

Academic Editor

PLOS ONE

Journal Requirements:

4. Please note that according to our submission guidelines (http://journals.plos.org/plosone/s/submission-guidelines), outmoded terms and potentially stigmatizing labels should be changed to more current, acceptable terminology. For example: “Caucasian” should be changed to “white” or “of [Western] European descent” (as appropriate).

5. You indicated that you had ethical approval for your study. In your Methods section, please ensure you have also stated whether you obtained consent from parents or guardians of the minors included in the study or whether the research ethics committee or IRB specifically waived the need for their consent.

Reviewers' comments:

Reviewer's Responses to Questions

**Comments to the Author**

1. Is the manuscript technically sound, and do the data support the conclusions?

Reviewer #1: Yes

Reviewer #2: Partly

2. Has the statistical analysis been performed appropriately and rigorously? 

Reviewer #1: Yes

Reviewer #2: No

3. Have the authors made all data underlying the findings in their manuscript fully available?

Reviewer #1: No

Reviewer #2: No

4. Is the manuscript presented in an intelligible fashion and written in standard English?

Reviewer #1: Yes

Reviewer #2: Yes

5. Review Comments to the Author

Reviewer #1: This paper reports findings on the interplay of relatives’ expressed emotions (EE), patients’ perceptions thereof, patients’ self-esteem and positive symptoms in the early stages of psychosis (i.e., in at-risk mental state (ARMS) and first-episode psychosis (FEP) participants). This is a well written manuscript. The introduction provides a coherent overview of the relevant literature, the research questions are clear, the methods and results are well structured, and the unique contributions to our understanding of EE, self-esteem and positive symptoms in early psychosis are well articulated. Overall, the paper is of interest. However, the paper could be improved further by addressing the following points, which are listed here in approximate order of appearance in the manuscript:

1. The abstract needs to explicitly state the aims of the study.

2. From reading the aims of the study (p. 4, line 90), it is not clear in which population these were addressed. For the second goal, an ‘early psychosis sample’ is mentioned, but more detail needs to be provided. Presumably ‘patients’ are individuals with ARMS and FEP, but this is not explicitly stated. What was the rationale for considering paranoia separately (thereby increasing the number of hypotheses even further)?

3. The section on “Participants and procedure” should more clearly focus on sample selection, recruitment, inclusion and exclusion criteria for patients and relatives. The actual sample size this yielded is commonly reported in the Results. The varying sample sizes are hard to follow – could Sample 1 and Sample 2 and their respective n be explicitly mentioned in all tables.

4. There are several methodological approaches to mediation analysis and, in particular, how the indirect effect was calculated. Which approach was adopted by the current paper?

5. The authors fitted an extremely high number of models to data from a relatively small sample – how did they ensure that these are robust and did not yield spurious findings? How did they account for multiple testing? The serial mediation models need to be described in more detail in the Method.

6. The reported indirect effects are of very small magnitude. This begs the question whether and to what extent these are clinically meaningful.

Reviewer #2: The topic of the current study (which examines the effects of expressed emotion and family environment) is interesting and there is a strong theoretical basis for the hypotheses outlined. However, there are an inordinate number of analyses conducted in a small sample, moreover, these are complex mediation analyses which have been conducted without first describing the underlying (simple) relationships. The result is that very hard to follow what has been done and whether this was justified. I would suggest the authors take a more strategic approach, rely less on 'black box' mediation analysis packages, and reduce the number of tests.

Specific comments:

1) Abstract: Can the authors report the specific groups examined (i.e., FEP/ARMS) - incipient psychosis is unclear.

2) Abstract is difficult to follow - only describes mediation models but does not first present the main analysis and so the nature of relationship between the exposure and outcome is unclear. (Although this is in fact a problem in the main text not just here).

3) Introduction (first paragraph) notes the importance of SE from a theoretical/hypothesised perspective but does not actually cite any studies/systematic reviews showing that self-esteem deficits are present in psychosis and ARMS. This is important to demonstrate as evidence for this is not particularly strong.

4) Introduction, line 53: EE does not need to be in the context of having an ill family member (even though it is a term that is more commonly seen in the psychosis literature). I think this sentence could be amended to be more accurate.

5) Line 55: 'early psychosis' is this first-episode (i.e., of full- threshold disorder) or early as in prodrome/CHR?

6) Figures 1-5 are helpful as the models/goals are complex and hard to follow - however, 5 separate figures seems far too much -It would be much better to combine these and this would also enable comparison. Although I would argue not all of these analyses are needed.

7) Methods: ARMS inclusion groups should at least be described briefly in the methods. Also better to note in the introduction a little bit about this group (at least that they are predominately characterised by attenuated psychotic symptoms and the proportion likely to transition). Were ARMS participants help-seeking?

8) Results: Important to present at least some information on sample characteristics in the main text - factors such as age, duration of illness (for FEP), sex/gender, will be important factors. Also useful to note who the relatives were in the text (i.e., vast majority mothers).

9) First line of indirect results, the correlations tables S5 and S6 show no relationships between relative's EE and symptoms. Surely then there is no need to proceed to mediation analyses because there is nothing to mediate? Step 1 (Baron & Kenny) is to regress the DV on IV to confirm a relationship but this step is absent?

10) Indirect effects of relative's EE on SE: same as above, table S8 shows no relationship between relative's EE and patient's SE so why advance to the next stage?

11) None of the models appear to account for other covariates that might confound/explain effects observed.

12) Discussion, first paragraph: The wealth and complexity of analyses means that it is difficult to determine whether the author's conclusions in this first paragraph are indeed accurate.

13) The suggestion that negative SE may be relevant for the 'development' of psychosis (line 364) is over-statement given that there are no longitudinal data and so can only say that these are correlated.

14) Discussion in general is long - perhaps better to save the discussion of moderating effects to a single paragraph rather than addressing after each finding.

15) Two biggest limitations not addressed: 1) sample size and 2) multiple testing, the number of tests performed is extraordinary high and very few are significant - those that are significant may be by chance and/or may reflect other confounding factors (e.g., age/sex).

6. PLOS authors have the option to publish the peer review history of their article (what does this mean?). If published, this will include your full peer review and any attached files.

Reviewer #1: No

Reviewer #2: No

---

## [Author Response · Author response to Decision Letter 0]

25 Feb 2021

Response to Editor’s comments

Thank you for submitting your manuscript to PLOS ONE. After careful consideration, we feel that it has merit but does not fully meet PLOS ONE’s publication criteria as it currently stands. Therefore, we invite you to submit a revised version of the manuscript that addresses the points raised during the review process.

Thank you for the constructive reviews of our manuscript, “The impact of family environment on self-esteem and symptoms in early psychosis”. We are delighted to have the opportunity to submit a revised manuscript, and we have revised the manuscript according to the Editor and Reviewers’ recommendations. 

We want to note that we identified a mistake in the Table S3 that presents the descriptive data on socio-demographic characteristics of early psychosis patients and their respective relatives. We really apologize for this mistake in the original submission. The mistake was located. Specifically, at the bottom of Table S3 it was reported that for Sample 2 the corresponding values for fathers were n=47 (81.0%) and for mothers n=5 (8.7%), whereas the correct numbers are the reverse, that is, n=47 (81.0%) for mothers and n=5 (8.7%) for fathers. It has now been corrected in the revised manuscript (in the Supporting Information file, S3 Table).

The manuscript has been revised following PLOS ONE's style requirements and we have now updated the file naming.

The authors of the present study confirm that some access restrictions apply to the data. The consent form that participants signed before participating in the study, approved by the Ethics Committee of the Universitat Autònoma de Barcelona (Comissió d'Ètica en l'Experimentació Animal i Humana (CEEAH); number 2697; http://www.recerca.uab.es/ceeah/), imposes restrictions for making the data publicly available. Participants agreed for all the data collected to be available to the members of the research group “Person-Environment Interaction in Psychopathology” led by Prof. Neus Barrantes-Vidal (Address: Departament de Psicologia Clínica i de la Salut, Facultat de Psicologia, Edifici B, Universitat Autònoma de Barcelona, 08193 Cerdanyola del Vallès, Spain; telephone: +34 93 5813864; email: neus.barrantes@uab.cat). This is the reason why data are available on request (requests should be addressed to the contact details provided above). 

We have now included captions for Supporting Information files at the end of the manuscript, and we have updated in-text citations to match accordingly.

 4. Please note that according to our submission guidelines (http://journals.plos.org/plosone/s/submission-guidelines), outmoded terms and potentially stigmatizing labels should be changed to more current, acceptable terminology. For example: “Caucasian” should be changed to “white” or “of [Western] European descent” (as appropriate).

Thank you for noticing this, we have changed the term “Caucasian-white” to “Western Europeans” in S3 Table.

 5. You indicated that you had ethical approval for your study. In your Methods section, please ensure you have also stated whether you obtained consent from parents or guardians of the minors included in the study or whether the research ethics committee or IRB specifically waived the need for their consent.

We have now included this statement in the Methods section.

Page 7, lines 153-154:

 “Written informed consent was also obtained from parents of the minors included in the study.”

Responses to Reviewer 1’s comments

We appreciated Reviewer’s positive comments and the constructive suggestions addressed to improve the scientific quality of the manuscript.

1. The abstract needs to explicitly state the aims of the study.

We have now included in the abstract a description of the main objectives of the study.

Page 2, lines 28-32:

 “The main objectives of this study were to examine whether: (1) patients’ SE mediated the effect of relatives’ EE on patients’ positive symptoms and paranoia; (2) patients’ perceived EE mediated the effect of relatives’ EE on patients’ SE; (3) patients’ SE mediated between patients’ perceived EE and patients’ symptomatology; and (4) patients’ perceived EE and patients’ SE serially mediated the effect of relatives’ EE on patients’ positive symptoms and paranoia.”

2. From reading the aims of the study (p. 4, line 90), it is not clear in which population these were addressed. For the second goal, an ‘early psychosis sample’ is mentioned, but more detail needs to be provided. Presumably ‘patients’ are individuals with ARMS and FEP, but this is not explicitly stated. What was the rationale for considering paranoia separately (thereby increasing the number of hypotheses even further)?

Thank you for bringing this to our attention; we have now provided more details about the samples employed for testing the goals of the study.

Page 4, lines 99-106.

 “The first goal of the present study was to explore in a sample of patients with ARMS and FEP and their respective relatives whether patients’ SE dimensions (positive and negative SE) mediated the effect of relatives’ EE dimensions (criticism and EOI) on patients’ symptoms (positive symptoms and paranoia) (Fig 1A). We predicted that patients’ negative SE would mediate the association between relatives’ EE dimensions and patients’ symptoms. In the second goal, we tested the Barrowclough’s model [14] in an early psychosis sample (patients with ARMS and FEP) by investigating the mediating role of patients’ perceived EE (perceived criticism and EOI) between relatives’ reports of EE and patients’ SE dimensions (Fig 1B).”

Regarding the issue of considering paranoia as an outcome separately, please note that we aimed to examine whether the same family environment factors and psychological mechanisms implicated in the development of paranoia were also implicated in the development of positive symptoms other than paranoia due to two fundamental issues. First, there is still not a universal consensus about the structure of positive symptoms. Some studies consider that paranoia, one of the most prevalent and ubiquitous symptom in psychosis, is better represented as a separate dimension, differentiated from other positive symptoms, whereas others offer support for a combined ‘reality distortion’ approach including all positive symptoms (Peralta & Cuesta, 2001). Second, but there is scarce knowledge about whether the hypothesized mechanisms underlying the onset and development of paranoia in some theoretical models are also relevant in the causal pathway to other forms of positive symptoms. Thus, given the scarcity of data and clarity on this regard we found it relevant to examine our hypotheses separately for paranoia and the other positive symptoms. This strategy allowed us to test whether the interaction of expressed emotion and self-esteem would impact on paranoia as well as other positive symptoms. 

3. The section on “Participants and procedure” should more clearly focus on sample selection, recruitment, inclusion and exclusion criteria for patients and relatives. The actual sample size this yielded is commonly reported in the Results. The varying sample sizes are hard to follow – could Sample 1 and Sample 2 and their respective n be explicitly mentioned in all tables.

We apologise for the lack of clarity and have now expanded on sample selection, recruitment and inclusion exclusion criteria for participants in the “Participants and procedure” section, including new information.

Page 6, lines 141-171:

 “The present study is embedded in a larger longitudinal study carried out in four Mental Health Centers of Barcelona (Spain) conducting the Sant Pere Claver- Early Psychosis Program [30]. Early psychosis patients (ARMS and FEP participants) and their respective relatives were included. ARMS criteria were established based on the Comprehensive Assessment of At-Risk Mental States (CAARMS) [31]. The CAARMS identifies 3 different at- risk mental states groups: the vulnerability group, the APS group, and the BLIPS group. The vulnerability group identifies those individuals with a combination of a trait risk factor and a significant deterioration in social and occupational functioning. The APS group includes those individuals with attenuated psychotic symptoms that not reach threshold levels of psychosis. Finally, the BLIPS group identifies individuals with brief limited intermittent psychotic symptoms that resolved spontaneously without antipsychotic medication. All the ARMS patients were help-seeking individuals, but none of the ARMS patients met DSM-IV- TR criteria [32] for any psychotic disorder or affective disorder with psychotic symptoms. FEP patients met DSM-IV-TR criteria [32] for any psychotic disorder or affective disorder with psychotic symptoms and presented a first-episode of psychosis within the past two years. Mean duration of illness was 11 months (SD=8.3), 12 months (SD=8.1) and 12 months (SD=7.4) for Sample 1, Sample 2 and Sample 3, respectively. However, 2 patients reached a length of 29 months in Sample 1 and 1 patient reached a length of 29 months in Samples 2 and 3. Patient’s inclusion criteria were age between 14 and 40 years old and IQ ≥ 75. Exclusion criteria for patients were evidence of organically based psychosis and any previous psychotic episode that involved pharmacotherapy. Relatives were referred to the study by their respective affected family members (i.e., early psychosis patients). Patients were informed of the relatives’ study and asked to name the person to whom they have a significant/close relationship. After getting the consent of the patient, the relative was contacted and was asked to participate into the study. Thus, the relatives recruited were those who had most regular contact and/or the most significant relationship with the patient. All participants provided written informed consent to participate and completed the assessment protocol within a maximum of 4 weeks. Written informed consent was also obtained from parents of the minors included in the study. The project was developed in accordance with the Code of Ethics of the World Medical Association (Declaration of Helsinki). Ethical approval was granted by the Ethics Committee of the Unió Catalana d’Hospitals (ref. 09–40) and by the Ethics Committee of the Universitat Autònoma de Barcelona (ref. 2679). All the interviews were conducted by experienced clinical psychologists. The time gap between patients and relatives’ assessments was minimal (range of 3 to 15 days).”

Following the Reviewer’s recommendation, we have now indicated in the heading of each table and supplementary table to which sample they correspond and the respective n (please see tables and supplementary tables). As suggested by the Reviewer, we have now reported the final sample size in the “Results” section instead of doing so in the “Participants and procedure” section.

Page 10, lines 233-250:

 “Depending on study goals, the samples examined differed (e.g., only patients or patient- relative dyads); therefore, there are different numbers of participants in the analyses. For goals examining patient-relative dyads (1, 2, 4 and 5), it was required that the examined measures had been responded by both members of the dyad. Hence, goal 1 included 77 patients (50 ARMS and 27 FEP; Sample 1) and their respective relatives, whereas Goals 2, 4 and 5 included 58 patients (37 ARMS and 21 FEP; Sample 2) and their respective relatives. For goals examining only patients (i.e., goal 3), 93 early psychosis patients (60 ARMS and 33 FEP; Sample 3) were included. Please note that a detailed participant flowchart is available in S1 Fig. As depicted in S1 Fig, there was a total sample of 122 relatives (n= 92 key relatives and n=30 second closest relatives) of early psychosis patients included at baseline. Taking into account that the present study focused on examining patient-relative dyads, only key relatives (n=92) were eligible as potential subjects of study. In samples of patient-relative dyads, relatives were mainly female [77.9% (Sample 1); 82.8% (Sample 2)], particularly patient’s mothers [75.3% (Sample 1); 81% (Sample 2)]. Mean age of the relatives was 50.71 years old (S.D = 10.8) and 50.69 years old (SD= 11.3) in Sample 1 and 2, respectively. Patients were predominantly male in all samples [70.1% (Sample 1); 72.4% (Sample 2); 68.8% (Sample 3)]. The mean age of the patients was 21.96 years old (S.D = 4.6), 22.05 years old (SD= 4.6) and 22.28 years old (SD= 4.4) in Samples 1, 2 and 3, respectively (please see Table S3 for details about relatives’ and patients’ socio- demographic characteristics).” 

4. There are several methodological approaches to mediation analysis and, in particular, how the indirect effect was calculated. Which approach was adopted by the current paper?

The methodological approach to mediation analysis used in the present manuscript is the Hayes’ method for assessing indirect pathways (Hayes, 2013, 2018). Please note that the statistical analysis section of the manuscript (pages 8-9, lines 203 to 229) references these procedures and the PROCESS macro (specifically, Andrew Hayes’ PROCESS macro) as follows: “Mediation analyses were performed using PROCESS v2.16 [44]”. The PROCESS macro was developed by Hayes (2013, 2018) on the basis of his own methodological approach for assessing indirect pathways. Please note that we believe that this approach follows best practices in the field as the study of mediation approaches has moved from traditional methods advocated by scholars such as Barron and Kenney (1986) that focused on change in the significance of the direct pathways when mediators were entered into the model to the specific examination of indirect pathways advocated by Hayes and others.

5. The authors fitted an extremely high number of models to data from a relatively small sample – how did they ensure that these are robust and did not yield spurious findings? How did they account for multiple testing? The serial mediation models need to be described in more detail in the Method.

We politely disagree with the Reviewer’s characterization of the analytic plan. We developed five specific questions/goals in the introduction and limited our analyses to these a priori hypothesized sets of analyses. We agree that spurious findings can be a concern, but we believe that this is much more the case in terms of post hoc “fishing expeditions.” Given the a priori nature of our planned analyses, we believe that post hoc p-value corrections would be inappropriate and unnecessarily raise the risk of Type II error. We thank the Reviewer’s suggestion to describe in more detail the serial mediation models in the method. Following the Reviewer’s recommendation, we have now included a more detailed explanation of the serial mediation models explored in the revised manuscript (page 9, lines 214 to 221):

“In the serial analysis, mediators are assumed to have a direct effect on each other [44], and the independent variable (relatives’ EE dimensions) is assumed to influence mediators (patients’ perceived EE and patients’ negative SE) in a serial way that ultimately influences the dependent variable (patients’ symptoms). Four different serial mediation models (SMM) were explored: [(SMM1: Relatives’ Criticism→ Perceived Criticism→ Negative SE→ Positive Symptoms); (SMM2: Relatives Criticism→ Perceived Criticism→ Negative SE→ Paranoia); (SMM3: Relatives’ EOI→ Perceived EOI→ Negative SE→ Positive Symptoms); (SMM4: Relatives EOI→ Perceived EOI→ Negative SE→ Paranoia)].”

6. The reported indirect effects are of very small magnitude. This begs the question whether and to what extent these are clinically meaningful.

We presume that the Reviewer is referring to the effect size of the indirect effects. Hayes (2018) discusses several indices of effect sizes for indirect effects, but also notes numerous limitations of each method. Furthermore, it is not clear that there are meaningful benchmarks for these indices (consistent with Cohen’s traditional small, medium, and large effect sizes). However, it would not be entirely surprising that indirect effects are of relatively small magnitude given that they often are tapping subtle processes that in the present case involve the interplay of the experiences of relatives and patients in the production of clinical and subclinical symptoms. Furthermore, effect size is an index of the magnitude of the association not the importance – and small effects often indicate meaningful patterns and pathways in human behaviour. There relatively smaller magnitude often represents the challenges of looking for these pathways and patterns amidst the “noisiness” of human behaviour. The clinical meaningfulness seems to be an issue of construct validation. This study completed a first step in this process by making and testing a priori hypotheses and identifying these likely small effects. Subsequent studies can continue this process of construct validation by further examining the specific contributions of these indirect effects to specific patterns of symptoms and impairment.

Responses to Reviewer 2’s comments

Reviewer #2: The topic of the current study (which examines the effects of expressed emotion and family environment) is interesting and there is a strong theoretical basis for the hypotheses outlined. However, there are an inordinate number of analyses conducted in a small sample, moreover, these are complex mediation analyses which have been conducted without first describing the underlying (simple) relationships. The result is that very hard to follow what has been done and whether this was justified. I would suggest the authors take a more strategic approach, rely less on 'black box' mediation analysis packages, and reduce the number of tests.

Thank you for the constructive review of our manuscript and the positive comments on the topic and theoretical basis of the study. We have revised the manuscript and discuss some of the suggestions offered below.

Specific comments:

1. Abstract: Can the authors report the specific groups examined (i.e., FEP/ARMS) - incipient psychosis is unclear.

We have now specified in the abstract the groups examined in the study.

Page 1, lines 34-63:

 “Incipient psychosis patients (at-risk mental states and first-episode of psychosis) and their respective relatives completed measures of EE, SE, and symptoms.”

2. Abstract is difficult to follow - only describes mediation models but does not first present the main analysis and so the nature of relationship between the exposure and outcome is unclear. (Although this is in fact a problem in the main text not just here).

Thank you for bringing this to our attention. We have changed the introductory part of the Abstract to make more understandable the nature of the relationship between the exposure and the outcome. We have also included a brief description of the aims of the study and we have eliminated a non-essential part of the Abstract in order to make it easier and more understandable.

Page 2, lines 25-43:

 “Expressed emotion (EE) and self-esteem (SE) have been implicated in the onset and development of paranoia and positive symptoms of psychosis. However, the impact of EE on patients’ SE and ultimately on symptoms in the early stages of psychosis is still not fully understood. The main objectives of this study were to examine whether: (1) patients’ SE mediated the effect of relatives’ EE on patients’ positive symptoms and paranoia; (2) patients’ perceived EE mediated the effect of relatives’ EE on patients’ SE; (3) patients’ SE mediated between patients’ perceived EE and patients’ symptomatology; and (4) patients’ perceived EE and patients’ SE serially mediated the effect of relatives’ EE on patients’ positive symptoms and paranoia. Incipient psychosis patients (at-risk mental states and first- episode of psychosis) and their respective relatives completed measures of EE, SE, and symptoms. Findings indicated that: (1) patients’ perceived EE mediated the link between relatives’ EE and patients’ negative, but not positive, SE; (2) patients’ negative SE mediated the effect of patients’ perceived EE on positive symptoms and paranoia; (3) the association of relatives’ EE with positive symptoms and paranoia was serially mediated by an increased level of patients’ perceived EE leading to increases in negative SE; (4) high levels of patients’ distress moderated the effect of relatives’ EE on symptoms through patients’ perceived EE and negative SE. Findings emphasize that patients’ SE is relevant for understanding how microsocial environmental factors impact formation and expression of positive symptoms and paranoia in early psychosis. They suggest that broader interventions for patients and their relatives aiming at improving family dynamics might also improve patients’ negative SE and symptoms”.

3. Introduction (first paragraph) notes the importance of SE from a theoretical/hypothesised perspective but does not actually cite any studies/systematic reviews showing that self-esteem deficits are present in psychosis and ARMS. This is important to demonstrate as evidence for this is not particularly strong.

As suggested by the Reviewer, we have cited several studies and systematic reviews where it is shown that self-esteem deficits are present across the whole psychosis continuum.

Page 3, line 49-50:

 “Recent research has demonstrated that low self-esteem is related to paranoia and positive symptoms across different stages of the psychosis continuum [4-6].”

4. Introduction, line 53: EE does not need to be in the context of having an ill family member (even though it is a term that is more commonly seen in the psychosis literature). I think this sentence could be amended to be more accurate.

We agree with the Reviewer that EE is a phenomenon that indeed may occur in any given family, irrespective of there being a diagnosis of a mental disorder. However, as the Reviewer notes, in the scope of clinical psychology and psychiatry EE was first described by Brown, Birley, and Wing (1972) to refer to relatives’ negative attitudes towards a family member with a mental disorder. We have now made this sentence more accurate.

Page 3, lines 56- 57:

 “Expressed emotion (EE) in psychiatry [7] is a measure of family emotional climate used to describe relatives’ attitudes towards a family member with a mental disorder.”

5. Line 55: 'early psychosis' is this first-episode (i.e., of full- threshold disorder) or early as in prodrome/CHR?

With the term early psychosis we wanted to refer to both first-episode of psychosis and the putative prodrome (CHR). However, in order to make the sentence more accurate we have changed its wording:

Page 3, 57-60:

 “The presence of high-EE attitudes [i.e., criticism and emotional over-involvement (EOI)] in families is related with poorer clinical outcome in chronic [8-10], first-episode of psychosis [11], and at-risk for psychosis patients [12].” 

6. Figures 1-5 are helpful as the models/goals are complex and hard to follow - however, 5 separate figures seems far too much -It would be much better to combine these and this would also enable comparison. Although I would argue not all of these analyses are needed.

Thank you for making us realize this point. We have now combined all the figures into a single figure (Figure 1).

Concerning whether all analyses were needed, we found it relevant to perform them for two main reasons. Firstly, most analyses are testing previous models proposed in the literature and thus have a priori hypotheses. Importantly, this study is testing them for the first time in an early psychosis sample comprising participants with first-episode psychosis and at-risk mental states for psychosis. Secondly, each of the mediation models proposed test the different individual hypotheses offered and, at the end, a serial mediation model articulates the theory-driven complex on the association between family environment and positive symptoms of psychosis. In summary, previous analysis (goals 1, 2, and 3) are the logical steps that aim to justify the final serial mediation model presented in this study (goal 4).

7. Methods: ARMS inclusion groups should at least be described briefly in the methods. Also better to note in the introduction a little bit about this group (at least that they are predominately characterised by attenuated psychotic symptoms and the proportion likely to transition). Were ARMS participants help-seeking?

We have now described ARMS inclusion groups in the Methods and we have added new information regarding ARMS recruitment.

Pages 6-7, line 145-158:

 “The CAARMS identifies 3 different at-risk mental states groups: the vulnerability group, the APS group, and the BLIPS group. The vulnerability group identifies those individuals with a combination of a trait risk factor and a significant deterioration in social and occupational functioning. The APS group includes those individuals with attenuated psychotic symptoms that not reach threshold levels of psychosis. Finally, the BLIPS group identifies individuals with brief limited intermittent psychotic symptoms that resolved spontaneously without antipsychotic medication. All the ARMS patients were help-seeking individuals, but none of the ARMS patients met DSM-IV-TR criteria [28] for any psychotic disorder or affective disorder with psychotic symptoms. (…) Patient’s inclusion criteria were age between 14 and 40 years old and IQ ≥ 75.” 

As suggested by the Reviewer we have included in the introduction more information about the definition and transitions rates of ARMS participants.

Page 4, lines 93-98:

 “ARMS individuals are predominately characterized by being young help-seeking individuals who experience attenuated positive psychotic symptoms that not reach threshold levels of psychosis. The transition risk to full-blown psychosis is around 22% at 3 years [25]; being severity of attenuated positive and negative symptoms as well as low functioning the most relevant factors associated with an increased risk [26].”

8. Results: Important to present at least some information on sample characteristics in the main text - factors such as age, duration of illness (for FEP), sex/gender, will be important factors. Also useful to note who the relatives were in the text (i.e., vast majority mothers).

Following the Reviewer’s recommendation, we have now provided detailed information on sample characteristics in the main text of the results’ section (page 10, lines 244 to 250):

 “In samples of patient-relative dyads, relatives were mainly female [77.9% (Sample 1); 82.8% (Sample 2)], particularly patient’s mothers [75.3% (Sample 1); 81% (Sample 2)]. Mean age of the relatives was 50.71 years old (S.D = 10.8) and 50.69 years old (SD= 11.3) in Sample 1 and 2, respectively. Patients were predominantly male in all the samples [70.1% (Sample 1); 72.4% (Sample 2); 68.8% (Sample 3)]. The mean age of the patients was 21.96 years old (S.D = 4.6), 22.05 years old (SD= 4.6) and 22.28 years old (SD= 4.4) in Sample 1, 2 and 3, respectively (please see Table S3 for details about relatives’ and patients’ socio- demographic characteristics).”

Moreover, following the Reviewer’s suggestion we have now provided more information about the inclusion criteria of FEP participants and we have indicated the duration of illness for FEP patients in all the examined samples (page 7, lines 152 to 157) as follows:

 “FEP patients met DSM-IV-TR criteria [32] for any psychotic disorder or affective disorder with psychotic symptoms and presented a first-episode of psychosis within the past two years. Mean duration of illness was 11 months (SD=8.3), 12 months (SD=8.1) and 12 (SD=7.4) for Sample 1, Sample 2 and Sample 3, respectively. However, 2 patients reached a length of 29 months in Sample 1 and 1 patient reached a length of 29 months in Sample 2 and 3.”

9. First line of indirect results, the correlations tables S5 and S6 show no relationships between relative's EE and symptoms. Surely then there is no need to proceed to mediation analyses because there is nothing to mediate? Step 1 (Baron & Kenny) is to regress the DV on IV to confirm a relationship but this step is absent?

10. Indirect effects of relative's EE on SE: same as above, table S8 shows no relationship between relative's EE and patient's SE so why advance to the next stage?

The questions above will be taken together. 

However, this is not the case for current approaches (such as Hayes) that examine indirect effects. 

The Reviewer is absolutely correct that following Baron and Kenny, the lack of a direct effect would preclude testing for mediation. However, the methodological approach to mediation analysis used in the present manuscript is the Hayes’ method for assessing indirect pathways (Hayes, 2013, 2018) and not the causal steps approach popularized by Baron and Kenny (1986). Nowadays, a statistically significant association between X and Y is not used as a prerequisite to searching for evidence of mediation. In the same way, the significance of path a (X on M) and path b (M on Y controlling for X) are not requirements to support a claim of mediation. Hayes (2009) suggested that new analytical opportunities arise if we quantify indirect effects rather than infer their existence from a set of tests on their constituent paths. In fact, in an email exchange with Professor Hayes, he specifically stated that significant associations of the IV and DV, IV and mediator, and mediator and DV “are not requirements of mediation analysis in the 21st century.” He added that a significant indirect effect is “all that matters with respect to whether X's effect is mediated by Y.”

11. None of the models appear to account for other covariates that might confound/explain effects observed.

We appreciate the Reviewer’s point, but we limited our analyses to a priori hypothesized indirect pathways. It is possible that other covariates might “confound/explain” the observed effects. However, such covariates were not the focus of our a priori hypothesized analyses. Furthermore, post hoc examination of such variables runs the risk of an exploratory fishing expedition, which would raise the risk of Type I error (see also our response to Reviewer 1, point # 5).

12. Discussion, first paragraph: The wealth and complexity of analyses means that it is difficult to determine whether the author's conclusions in this first paragraph are indeed accurate.

We appreciate the Reviewer’s opinion. However, we sincerely consider that the first paragraph of the discussion only describes the findings of the study in the context of published literature. We respectfully think that the text in the first paragraph is thoroughly based on the specific results obtained in the present study, and therefore it is accurate. Nevertheless, we have amended the last sentence of the first paragraph (as suggested by the reviewer in the comment # 13 below) in order to avoid an overstatement of specific results. 

13. The suggestion that negative SE may be relevant for the 'development' of psychosis (line 364) is over-statement given that there are no longitudinal data and so can only say that these are correlated.

Following the Reviewer’s observation, we have amended the sentence as follows (page 23, lines 397- 400):

 “Our findings also revealed that negative, but not positive, SE was the most common mediating factor between EE and symptoms, suggesting that negative SE may be especially related to positive symptoms and paranoia [46], and highlighting the significance of separately exploring positive and negative SE [17, 36, 37]. ”  

14. Discussion in general is long - perhaps better to save the discussion of moderating effects to a single paragraph rather than addressing after each finding.

We have now combined the discussion of the moderating role of group into a single section rather than addressing this issue after each specific finding. Moreover, we have eliminated some redundant and non-essential parts in order to make it shorter and easy to read. Please, see “The moderating role of group” section on page 23.

15. Two biggest limitations not addressed: 1) sample size and 2) multiple testing, the number of tests performed is extraordinary high and very few are significant - those that are significant may be by chance and/or may reflect other confounding factors (e.g., age/sex).

Thank you for bringing this to our attention. We agree that due to limitations in our sample size some of the statistically significant results may be due to chance and/or may reflect other confounding factors, as occurs in most studies with relatively small samples sizes. We have included a new statement noticing the limitations of our sample size (Page 26, lines 536-537)

 “Due to limitations on the sample size, findings and conclusions from the present study must be interpreted carefully.”

Regarding the concern about multiple testing, please note that we developed five specific goals in the introduction and limited our analyses to test these a priori hypotheses. Regarding indirect effects, 8 of the 18 mediation models tested were significant, whereas 7 of the 10 non-significant models were those including the positive dimension of self-esteem. This seems to indicate that negative self-esteem is more relevant than positive self-esteem in the link between EE and positive symptoms, and the statistically non-significant results inform us on the different role of positive and negative SE. We also agree that spurious findings can be a concern, but think that this would actually be more so the case in terms of post hoc “fishing expeditions.” Given the a priori nature of our planned analyses, we believe that post hoc p-value corrections would be inappropriate and unnecessarily raise the risk of Type II error (please see also point #11).

References

Baron, R. M., & Kenny, D. A. (1986). The moderator–mediator variable distinction in social psychological research: Conceptual, strategic, and statistical considerations. Journal of Personality and Social Psychology, 51(6), 1173–1182. https://doi.org/10.1037/0022-3514.51.6.1173

Hayes, A. F. (2013). Introduction to mediation, moderation, and conditional process analysis: A regression-based approach. (1st ed.). The Guilford Press.

Hayes, A. F. (2018). Introduction to mediation, moderation, and conditional process analysis: A regression-based approach. (2nd ed.). The Guilford Press.

Peralta, V., & Cuesta, M. J. (2001). How many and which are the psychopathological dimensions in schizophrenia? Issues influencing their ascertainment. Schizophrenia research, 49(3), 269–285. https://doi.org/10.1016/s0920-9964(00)00071-2

---

## [Decision Letter · Decision Letter 1]

24 Mar 2021

The impact of family environment on self-esteem and symptoms in early psychosis

PONE-D-20-31530R1

Dear Dr. Barrantes-Vidal,

We’re pleased to inform you that your manuscript has been judged scientifically suitable for publication and will be formally accepted for publication once it meets all outstanding technical requirements.

Kind regards,

Therese van Amelsvoort

Academic Editor

PLOS ONE

Additional Editor Comments (optional):

Reviewers' comments:

Reviewer's Responses to Questions

**Comments to the Author**

1. If the authors have adequately addressed your comments raised in a previous round of review and you feel that this manuscript is now acceptable for publication, you may indicate that here to bypass the “Comments to the Author” section, enter your conflict of interest statement in the “Confidential to Editor” section, and submit your "Accept" recommendation.

Reviewer #1: All comments have been addressed

Reviewer #2: All comments have been addressed

2. Is the manuscript technically sound, and do the data support the conclusions?

Reviewer #1: Partly

Reviewer #2: Partly

3. Has the statistical analysis been performed appropriately and rigorously? 

Reviewer #1: I Don't Know

Reviewer #2: Yes

4. Have the authors made all data underlying the findings in their manuscript fully available?

Reviewer #1: Yes

Reviewer #2: No

5. Is the manuscript presented in an intelligible fashion and written in standard English?

Reviewer #1: Yes

Reviewer #2: No

6. Review Comments to the Author

Reviewer #1: (No Response)

Reviewer #2: (No Response)

7. PLOS authors have the option to publish the peer review history of their article (what does this mean?). If published, this will include your full peer review and any attached files.

Reviewer #1: No

Reviewer #2: No

---

## [Editor Report · Acceptance letter]

26 Mar 2021

PONE-D-20-31530R1 

The impact of family environment on self-esteem and symptoms in early psychosis 

Dear Dr. Barrantes-Vidal:

I'm pleased to inform you that your manuscript has been deemed suitable for publication in PLOS ONE. Congratulations! Your manuscript is now with our production department. 

Kind regards, 

on behalf of

Prof. Therese van Amelsvoort 

Academic Editor

PLOS ONE